# Waist Circumference and All-Cause Mortality among Older Adults in Rural Indonesia

**DOI:** 10.3390/ijerph16010116

**Published:** 2019-01-03

**Authors:** Cahya Utamie Pujilestari, Lennarth Nyström, Margareta Norberg, Nawi Ng

**Affiliations:** Department of Epidemiology and Global Health, Faculty of Medicine, Umeå University, 90187 Umeå, Sweden; lennarth.nystrom@umu.se (L.N.); margareta.norberg@umu.se (M.N.); nawi.ng@umu.se (N.N.)

**Keywords:** abdominal obesity, deaths, Indonesia, older people, waist circumference

## Abstract

Waist circumference, a measure of abdominal obesity, is associated with all-cause mortality in general adult population. However, the link between abdominal obesity with all-cause mortality in the studies of older adults is unclear. This study aims to determine the association between waist circumference and all-cause mortality in older adults in Indonesia. The association between waist circumference and all-cause mortality was examined in 10,997 men and women aged 50 years and older, in the World Health Organization (WHO) and International Network of field sites for continuous Demographic Evaluation of Populations and their Health in developing countries (INDEPTH) collaboration Study on global AGEing and adult health (SAGE) in Purworejo District Central Java, Indonesia during 2007–2010. Multivariate Cox regression analysis with restricted cubic splines was used to assess the non-linear association between waist circumference and all-cause mortality. During the 3-year follow-up, a total of 511 men and 470 women died. The hazard ratio plot shows a pattern of U-shape relationship between waist circumference and all-cause mortality among rich women, though the result was significant only for women in the lower end of waist circumference distribution (*p* < 0.05). Poor men with a low waist circumference (5th percentile) have a two times higher mortality risk (HR = 2.1; 95% CI = 1.3, 3.3) relative to those with a waist circumference of 90 cm. Poor women with a low waist circumference (25th percentile) have a 1.4 times higher mortality risk (HR = 1.4; 95% CI = 1.1, 1.8) relative to those with a waist circumference of 80 cm. This study shows a significant association between low waist circumference measure and mortality, particularly among poor men and women. Though the association between large waist circumference and mortality was not significant, we observed a trend of higher mortality risk particularly among rich women with large waist circumference measure. Public health intervention should include efforts to improve nutritional status among older people and promoting healthy lifestyle behaviours including healthy food and active lifestyle.

## 1. Background

Obesity is one of the main risk factors for chronic diseases. Worldwide, the prevalence of obesity has nearly doubled between 1980 and 2014 [1]. The World Health Organization (WHO) has estimated that in 2014, 11% of men and 15% of women aged 18 years and above were obese while another 38% of men and 40% of women were overweight [1]. With regard to the general adult population, obesity is known to be associated with many health problems such as cardiovascular diseases (CVDs), diabetes, certain cancers [1], as well as premature deaths [2], but for the older population, the associations are less clear [3,4,5,6,7,8,9].

With the increase of overweight and obesity in the older population [10], an assessment of the associated health problems will be necessary in order to develop prevention strategies. Although most studies have found that obesity was a strong risk factor for morbidity and mortality in the general adult population, the relationship has been inconsistent in the older adult population [8,11]. Many studies have observed a decreased mortality rate in older adults with obesity or who are overweight compared to those of a healthy weight [7,8,12]. This phenomenon, namely the ‘obesity paradox’, has been reported not only in older adults with specific chronic conditions but also among community-dwelling older adults [8,13,14].

Body mass index (BMI), calculated as weight in kilogrammes divided by height in metres squared, and, to a lesser extent, waist circumference, are widely used in clinical practice as measures to assess a link between obesity and an individual’s health risk [3,4,5,6,7,8,9]. However, many studies have shown that using BMI to identify obesity among older adults may be inaccurate due to the loss of lean muscle mass and increase in visceral body fat [15]. Vlassopoulos et al. showed that after the age of 65 the BMI tended to decrease [16], but decreasing BMI among older adults does not imply that excess fat storage is uncommon. Therefore, many have recommended the use of waist circumference as a better means of identifying obesity among older adults [17,18,19,20].

In Indonesia, obesity has been increasing dramatically at all ages over the past decades [21], and so has its impact on chronic disease [22] and mortality [1]. The association between obesity and mortality remains to be determined, especially in Indonesian older adults. Therefore, more research among the older population is needed as it would provide an evidence base in the area. This study examined the association between waist circumference and mortality among a population aged 50 years and older in Purworejo district, Central Java, Indonesia. We also examined if the association differ between men and women and between individuals in different socioeconomic groups.

## 2. Methods

### Study Setting

The WHO-INDEPTH longitudinal Study on global AGEing and adult health (SAGE) of people aged 50 years and older in Purworejo District, Central Java Province, Indonesia was conducted in 2007. Purworejo district is located in a Southern Java Island with a population of 712,686 inhabitants on an area of 1035 km^2^ [23]. A Health and Demographic Surveillance System (HDSS) site was established in Purworejo District in 1994, covering a total of 55,000 individuals in 14,500 households. Within the site, individual demographic data (birth, death, marital status, migration, etc.) were collected annually, while household socioeconomic data was collected every 5th year. The HDSS site has been a basis for many studies embedded in the surveillance site, including the WHO-INDEPTH SAGE [24].

In 2007, a total of 11,753 older adults aged 50 years and older were recruited to the WHO-INDEPTH SAGE. These individuals were followed up in 2010, among whom 1199 had died and 1033 were categorised as lost to follow-up (i.e., 59 refused to participate, 176 were not found at home after three visits, 575 were out-migrated and a further 223 could not be interviewed due to different reasons). Of these 11,753 individuals, information about date of death was not available for 152 individuals and a total of 604 respondents had missing data on other key variables included in this study (i.e., waist circumference, age, occupation, wealth status). Thus, 10,997 respondents with complete data were included in the longitudinal analysis (Figure 1).

## 3. Instruments and Variables

We used the individual and household-level WHO-INDEPTH SAGE questionnaires. The questionnaires were administered at home through face-to-face interviews by a trained interviewer [25]. The individual questionnaire assessed respondents’ socio-demographic characteristics including age, sex, education, occupation, marital status, residence and self-reported chronic conditions. The household questionnaire assessed information on housing condition, infrastructure facilities, and ownership of assets. Following the interview, waist circumferences were measured at the point midway of the last palpable rib and the top of the iliac crest using inelastic measuring tape in centimetres (cm) [26].

The main outcome measure in this study was death, which was extracted from the routine HDSS surveillance database in 2010. Information about cause-of-death was not collected, hence in this study, we analysed all-cause mortality.

The main exposure of interest in this study was waist circumference. According to the WHO cut-off recommendation for the Asian populations, waist circumference ≥90 cm among men and ≥80 cm among women were considered as abdominal obesity [26,27].

Other potential confounders measured at the baseline in 2007 that were controlled in the analysis include age, sex, education, occupation, marital status, residence, self-reported chronic conditions and wealth status.

Age was grouped in ten-yearly intervals as 50–59, 60–69, 70–79 and 80+ years old. Education was categorised as ‘no-formal education’ (never having any formal education), ‘≤6 years’ (not completed elementary school, completed elementary school, not completed junior high school), ‘>6 years’ (completed junior high school, high school, academy or university, master’s degree). Occupation was categorised as ‘non-physical labour’ (those who worked as government workers, non-government workers or self-employed), ‘no occupation’ (those who were retired, housewives, or not-having a job), and ‘physical labour’ (those who worked as fishermen, farmers, labourer, or rickshaw drivers). Marital status was categorised as ‘single/widowed’ (not married, divorced, separated, widowed), ‘partnership’ (married, living together). Residential area was categorised based on respondent’s living environment which also differs socioeconomically as ‘inland’, ‘coastal’, ‘hilly’ and ‘mountainous’. Self-reported chronic diseases were based on respondents’ self-reporting on one or more chronic conditions including hypertension, diabetes, stroke, cardiovascular disease, and asthma.

Wealth status was created using principal component analysis (PCA) from selected household key assets. The households were categorised into poor and rich households based on the median value of the PCA scores [28]. More details about how the PCA was conducted have been published elsewhere [29].

## 4. Statistical Analyses

Baseline characteristics were described in number and percentage grouped by sex. Cumulative hazard curves in men and women were derived using the Nelson-Aalen cumulative hazard functions, stratified by abdominal obesity and wealth status. Time to event was calculated as the date of interview in the baseline through date of death during 2007–2010 and was estimated in person-months. Individuals who were followed-up in 2010, or outmigration, or lost-to-follow-up were censored in the analysis.

To investigate the association between waist circumference and all-cause mortality, hazard ratios (HR) and 95% confidence intervals (CIs) were calculated using Cox proportional hazards models.

In multivariable Cox proportional hazards models, waist circumference was adjusted for these variables: sex, age group, marital status, education, occupation, residence, self-reported chronic disease and wealth status. We also performed multivariable stratified analyses on sex and wealth status (men-poor, men-rich, women-poor, women-rich). These multivariable stratified analyses were performed by adjusting for the remaining variables.

To examine whether waist circumference should be modelled as linear or non-linear in the multivariable Cox regression analysis for mortality, we compared the Akaike Information Criterion (AIC) values (data not shown). The non-linear relationships between waist circumference and all-cause mortality were modelled using restricted cubic splines analysis, with 4 degrees of freedom [30]. All the models were adjusted. We ran the linear and non-linear model for both full and stratified analyses by sex and wealth status. The model using splines showed the lowest AIC score. Therefore, we decided to use the restricted cubic splines in the Cox regression analysis. For all analyses, *p* < 0.05 (two-tailed) was taken as the threshold for statistical significance which corresponds to the 95% confidence interval. For comparison purposes, the hazard ratios were extracted in the 5th, 25th, 75th and 95th percentiles of the waist circumferences, where risks were relative to the reference value for each sex. The reference values (men = 90 cm, women = 80 cm) were based on abdominal obesity cut-off value for Asian populations [26,27]. Lastly, the estimated HR were presented graphically with 95% CI bands across the whole waist circumference range.

All statistical analyses were performed using Stata Version 15.1 (StataCorp, College Station, TX, USA). We used RStudio version 1.0.153 (The R foundation for statistical computing, Boston, MA, USA) to plot the estimated HR and 95% CI.

## 5. Ethics

The ethical boards of the Faculty of Medicine, Gadjah Mada University, Indonesia, granted the ethical approval for the study in the Purworejo District (IRB number: KE/FK/69/EC). Information about the study was provided to each participant verbally prior to the data collection, followed by obtaining written informed consent. The participants were informed that they could withdraw at any stage of the study.

## 6. Results

A total of 5129 men and 5868 women were included in the analysis of whom 511 men and 470 women had died during the follow-up period (Table 1). The majority of men and women had fewer than 6 years of education (63% men and 49% women), were physical labourers (77% men and 59% women) and were in partnerships (87% men and 59% women).

The percentage of single and widowed was significantly higher in women (42% vs. 13%; *p* < 0.001), as was self-reported chronic disease (20% vs. 18%; *p* = 0.001). At baseline in 2007, abdominal obesity was seven times more prevalent among women as compared with men (37% vs. 6.3%; *p* < 0.001).

Figure 2 shows that poor obese men had a higher mortality rate compared to the other groups in the first 20 months of the follow-up period. At the end of the follow-up period, the mortality rate was higher among the rich obese and poor non-obese men. For women, steadily increasing curves across the follow-up period were observed, except for a considerable variation that was seen among the poor obese women from the first half to the second half of the follow-up period. The poor non-obese and rich non-obese showed a higher mortality rate, especially at the end of the follow-up period. Different pattern was observed when comparing men and women graph. The rich obese men had the highest mortality rate compared to the other groups, in contrast to the rich obese women who had the lowest.

Table 2 and Table 3 and Figure 3 show the results from the stratified Cox regression analysis with restricted cubic splines. Low waist circumference was positively associated with increased risk of mortality in men and women (Table 2). Poor men with a low waist circumference (5th percentiles) have a two times higher mortality risk relative to those with waist circumferences of 90 cm (HR = 2.1; 95% CI = 1.3, 3.3). Meanwhile, rich men in the 5th percentiles showed a nonsignificant association with mortality (HR = 1.0; 95% CI = 0.67, 1.5). There was no significant association observed among men with a large waist circumference.

Poor women with a low waist circumference (25th percentiles) had a 1.4 times higher mortality risk relative to those with a waist circumference of 80 cm (HR = 1.4; 95% CI = 1.1, 1.8). Although the mortality risk was higher, there was no significant association between waist circumference and mortality in rich women with a large waist circumference (95th percentile, HR = 1.2; 95% CI = 0.78, 2.0).

We conducted sensitivity analyses to address potential reverse causality in assessing the association between waist circumference and mortality (See Appendix A). The effects sizes of hazard ratio of either excluding study participants who reported chronic disease in the baseline (Appendix A) or excluding all deaths occurring within the first 24 months of the follow-up (Appendix A) are very consistent with the hazard ratio we reported in Table 2, despite some of the results turned insignificant. These additional analyses ensure the robustness of the results presented in this paper.

Overall, the hazard ratio increased with age (Table 3). Men and women with chronic disease had higher hazard ratios compared to those without any chronic disease, regardless of their wealth status. The hazard ratios were significantly higher among poor (HR = 3.9; 95% CI = 1.4, 10) and rich (HR = 2.0; 95% CI = 1.1, 3.8) women with no occupation, single/widowed rich women (HR = 1.6; 95% CI = 1.2, 2.2) and poor men who lived in the inland area (HR = 1.6; 95% CI = 1.2, 2.1).

Almost all plots of mortality hazard ratio showed wide confidence intervals, especially for individuals at both ends of the waist circumference measurement (Figure 3). Poor men with waist circumferences <70 cm had a significant increase in the mortality hazard ratio, relatively to those with waist circumference of 90 cm. No significant association between waist circumference and all-cause mortality was observed among poor men with waist circumferences above 90 cm. The association between waist circumference and all-cause mortality among rich men was, however, not statistically significant at all.

There was a tendency towards a U-shaped association between waist circumference and all-cause mortality in women, particularly among the rich women even though the confidence interval was wide, and the association was not significant. The estimates of mortality hazard ratio for waist circumference over 80 cm among poor women was below one, and no significant association was also observed. Poor women with a waist circumference less than 80 cm, however, had an increased mortality hazard ratio with a significant association observed within the range of waist circumference between 62 and 73 cm.

## 7. Discussion

The association between waist circumference and all-cause mortality was found to be U-shaped for women aged 50 years and older in Purworejo district. Though the association was not significant at large waist circumference, the U-shaped association indicates that both low and large waist circumference might have negative effect to the longevity of women, particularly wealthy women. Such association was not shown in men.

Reduced mortality risk was observed among poor men and women with large waist circumferences and increased mortality risk was observed for those with low waist circumferences. Among the poor, having a waist circumference larger than the abdominal obesity cut-offs might be interpreted as being protective to death. In line with our findings, the NHANES III study conducted in the US among adults aged 30–102 years old showed that higher waist circumference in men and higher BMI in men and women were associated with a lower mortality risk [7]. In contrast, the Cancer Prevention Study II Nutrition Cohort of men and women aged 50 years and older showed that a larger waist circumference was strongly associated with a higher risk of mortality [17].

The reverse relation of obesity and mortality risk in older adults has been challenged for several reasons. Namely selective survival (where obese younger and middle-aged have premature death, and the remaining subjects are the healthy obese subjects), the cohort-effects (the current older obese subjects have been exposed to different lifestyles and environments with fewer obesity risk factors) and the redistribution of body fat at older age [19]. Inadequate adjustment or consideration for intermediary factors in the causal pathway of waist circumference and mortality such as smoking and a pre-existing chronic disease which can modify the association was also mentioned as a contributing factor in the reverse obesity-mortality relationship [31]. In this study, although information on smoking status and doctor diagnosis on pre-existing chronic disease was not available, the relation between waist circumference and mortality was adjusted for several confounders which included information on the self-reported chronic diseases. Therefore, we believe that we were able to assess the association between waist circumference and mortality accurately.

A beneficial or neutral association of obesity on mortality among older people was previously documented in other studies, especially studies that used BMI as the obesity measurement [7,8,9]. Although BMI is commonly used to quantify obesity in population-based studies, the changes in body composition and decline in height due to the compression of vertebral bodies that occurs in old age, may register the BMI falsely. General obesity (measured with BMI) is less pronounced among older individuals, as after the age of 60 people tend to lose their weight but elevate their waist [16]. Studies using waist circumference tended to show a more precise association between obesity and mortality [3,4,29].

Researchers argue that the inconsistent relationship between obesity and mortality among older adults might be due to the differences in the obesity measurements, i.e., BMI and waist circumference, as illustrated in the two meta-analyses that examined the association using a different measurement. Winter et al. found that older individuals with a BMI in the overweight range had a lower mortality risk [8], while Hollander et al. found an increased mortality risk in older people with larger waist circumferences [4]. Taken together, the findings from previous studies have noted that a higher BMI or larger waist circumference among older adults had a protective effect that may outweigh the potential adverse effects. Being ‘moderately overweight’ may provide energy reserves during illness especially among the older and poorer populations [32,33].

Importantly, we observed that the all-cause mortality risk started to increase at a waist circumference of <90 cm in men or <80 cm in women, which should fall within the healthy waist circumference range as suggested by WHO for Asian populations [26,27]. This finding might suggest the need to lower the optimal cut-off values for the Indonesian population. To date, several studies have proposed specific cut-off values in a different age, sex and ethnic groups within Asian populations [34,35,36,37,38], including Indonesian [39]. The recent study on anthropometric cut-off values for obesity screening among Indonesian adults aged 18–65 years old proposed a waist circumference cut-off values of 76.8 cm and 71.7 cm for men and women, respectively [39]. Nevertheless, studies to investigate the optimal waist circumference cut-off value in a representative sample of the Indonesian population, which include a broader range of age groups, are needed to address inconsistencies in the literature.

Socioeconomic wellbeing is one crucial determinant of better health and it lowers mortality risk [40]. The English Longitudinal Study of Ageing reported that the wealthy tended to have better health and longer life compared to the poor [41]. In our study, obesity is more common among the rich [29] who also had a higher mortality risk than the poor obese. On the contrary, mortality risk among the poor with low waist circumference was notably higher than among their wealthier counterparts.

We hypothesised that the older adults with a very low waist circumference were sick. Better access and capability to seek health care allowed the rich to have early detection and proper management of the diseases [42] that might be attributable to the lower mortality risk among the wealthy group with low waist circumference compared to the poorer group. Our hypothesis was supported by a focus group discussion conducted in Purworejo among adults aged 25+ years old which mentioned that people would seek health care while they have the financial capability, and the poor who do not have that capability should be prepared to die [43].

This study also observed a different association of obesity with mortality between men and women, which might be attributable to the difference in the death rate and abdominal obesity prevalence. Gender difference in the abdominal obesity prevalence has been observed in the previous studies conducted in Indonesia [21,29,44] with physical inactivity as the prime contributing factor [45]. Additional studies relating to the disparity in the BMI of Indonesian men and women aged 20–59 years old also supported the finding that physical activity explains the significant part of the difference in men and women BMI [36]. Moreover, the study also noted that men and women who lived in different residential areas (e.g., urban or rural area) were engaged in different intensities of exercise [44].

Although the likelihood of chronic health problems increases with age, research showed that risk factors were established at a younger age. For example, osteoarthritis is associated with obesity in middle-age [46]. Alzheimer’s and dementia may also be the results of midlife obesity [47]. The NHANES III study suggest that measures of body fat distributions among the middle-aged are essential in identifying an increased risk of mortality in the older-aged [7]. This suggest that obesity prevention strategies should be started before middle age to avoid chronic health problems during older age.

It is also important to remember that the prevention strategies should not be limited to the wealthier populations who have higher obesity prevalence, because the burden of chronic diseases is detrimental to the poorer populations. Improving access to health information and health care for the more impoverished populations is very important to reduce inequality in health.

### Strength, Limitations and Future Studies

This study has several strengths. First, this panel study was conducted in a large representative sample of older adults in a well-established HDSS site with more than 90% follow-up rate. Despite showing non-significant association, this study which focuses on waist circumference and obesity among older people in Purworejo, Indonesia may contribute to filling the knowledge gap in the area.

The limitations of this study primarily concern the fact that the three-year follow-up time was not long enough to capture more death events. Studies assessing the association between obesity and mortality require large samples to capture the wide range of adiposity measurements as well as a sufficient follow-up period [4,8]. Ideally, the study should also have included obesity status at middle age to avoid the selective survival bias and the cohort effects [48].

Another limitation is that our death records do not contain cause-specific death, which may have been more accurate in predicting deaths related to abdominal obesity. It is also important to include BMI measurement to be able to conduct more analysis and compare the effect of BMI and waist circumference. The facts that our height and weight data were self-reported with high missing rates (23%) did not allow us to explore the association between BMI and mortality or to compare the association with waist circumference.

Future studies should consider a longer follow-up period not only to capture more events but also to clarify the potential influence of confounding such as smoking and undiagnosed disease. It is also important to collect information on smoking status and chronic disease diagnosis to be able to assess the impact of these confounders among the older population.

## 8. Conclusions

This study shows a significant association between low waist circumference measure and mortality, particularly among poor men and women. Though the association between large waist circumference and mortality was not significant, we observed a trend of higher mortality risk particularly among rich women with large waist circumference measure. Public health intervention should include efforts to improve nutritional status among older people and promoting healthy lifestyle behaviours including healthy food and active lifestyle.

## Figures and Tables

**Figure 1 ijerph-16-00116-f001:**
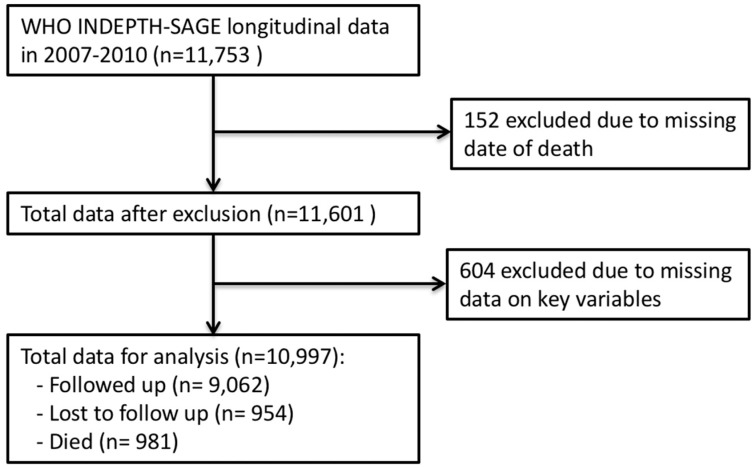
Flowchart of the study population generated from WHO-INDEPTH SAGE Purworejo longitudinal data.

**Figure 2 ijerph-16-00116-f002:**
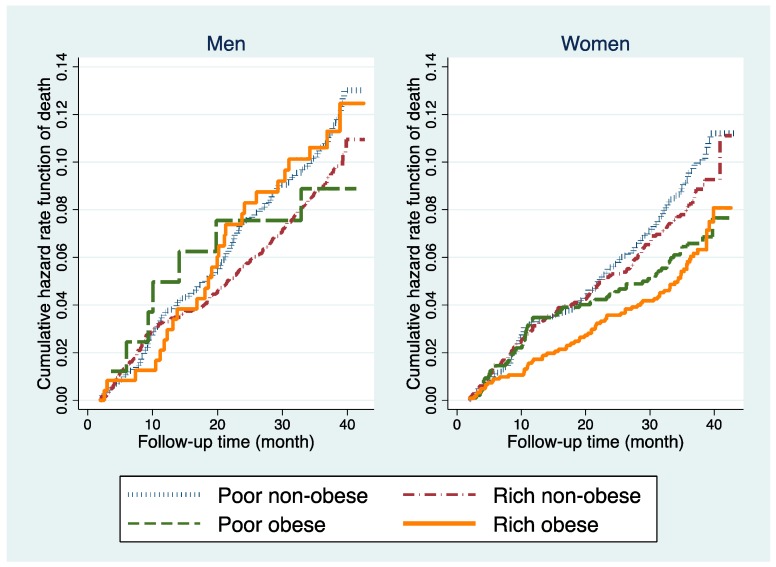
Nelson-Aalen cumulative hazard curve in men and women, stratified by wealth and abdominal obesity status during the study period (2007–2010). WHO-INDEPTH SAGE Purworejo longitudinal data.

**Figure 3 ijerph-16-00116-f003:**
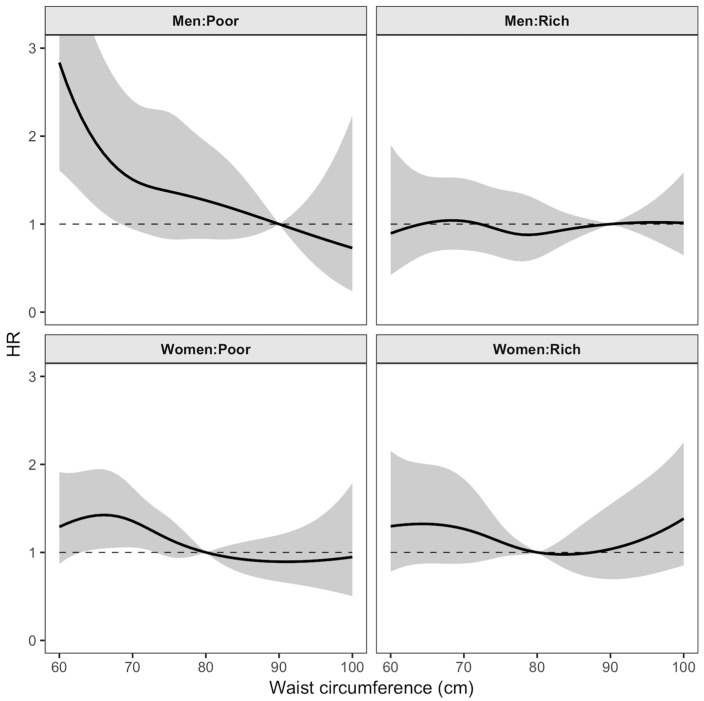
Plots of mortality hazard ratio (HR) with 95% confidence interval (shaded regions) from multivariate adjusted Cox regression analysis with restricted cubic splines of waist circumference. WHO-INDEPTH SAGE Purworejo longitudinal data (2007–2010).

**Table 1 ijerph-16-00116-t001:** Percentage (%) of men and women by socioeconomic characteristics and abdominal obesity at baseline in 2007 and death rate/1000 people during the study period 2007–2010 (n = 10,997). WHO-INDEPTH SAGE Purworejo longitudinal data.

Characteristic	Number (%)		Death Rate/1000 people	
	Men (n = 5129)	Women (n = 5868)	Men (n = 511)	Women (n = 470)
Sex			99	80
Age (years)				
50–59	1921 (37.5)	2130 (36.3)	31	27
60–69	1681 (32.8)	2097 (35.7)	89	74
70–79	1186 (23.1)	1331 (22.7)	172	137
80+	341 (6.6)	310 (5.3)	287	235
Education				
No formal education	803 (15.7)	2377 (40.5)	143	116
≤6 years	3217 (62.7)	2899 (49.4)	94	56
>6 years	1109 (21.6)	592 (10.1)	83	51
Occupation				
Non-physical labour	470 (9.1)	491 (8.4)	76	31
No occupation	701 (13.7)	1915 (32.6)	231	130
Physical labour	3958 (77.2)	3462 (59.0)	79	59
Marital status				
Single/widowed	648 (12.6)	2418 (41.2)	172	117
Partnership	4481 (87.4)	3450 (58.8)	89	54
Self-reported chronic disease				
No	4235 (82.6)	4710 (80.3)	77	65
Yes	894 (17.4)	1158 (19.7)	204	142
Residence				
Coastal	2556 (49.8)	2943 (50.2)	94	84
Inland	1225 (23.9)	1473 (25.1)	122	71
Hilly & mountainous	1348 (26.3)	1452 (24.7)	89	80
Wealth status				
Poor	2521 (49.2)	3141 (53.5)	107	86
Rich	2608 (50.8)	2727 (46.5)	92	74
Abdominal obesity	322 (6.3)	2198 (37.5)	102	63

**Table 2 ijerph-16-00116-t002:** All-cause mortality stratified by sex and wealth status. Hazard ratio (HR) and 95% confidence intervals (CI) from multivariate adjusted Cox regression analysis with restricted cubic splines of waist circumference at 5th, 25th, 75th and 95th percentiles. WHO-INDEPTH SAGE Purworejo longitudinal data (2007–2010).

	N	Percentiles							
		5th		25th		75th		95th	
		Waist (cm)	HR (95% CI)	Waist (cm)	HR (95% CI)	Waist (cm)	HR (95% CI)	Waist (cm)	HR (95% CI)
Men									
Poor	2521	64	2.06 (1.28, 3.31) *	70	1.51 (0.94, 2.41)	80	1.27 (0.83, 1.94)	88	1.06 (0.91, 1.22)
Rich	2608	65	1.01 (0.67, 1.54)	72	1.00 (0.69, 1.47)	83	0.92 (0.74, 1.16)	93	1.01 (0.92, 1.12)
Women									
Poor	3141	62	1.36 (0.97, 1.90)	69	1.39 (1.06,1.82) *	81	0.98 (0.93, 1.03)	92	0.89 (0.64, 1.25)
Rich	2727	63	1.32 (0.86, 2.03)	72	1.22 (0.89, 1.66)	85	0.98 (0.77, 1.24)	97	1.24 (0.78, 1.98)

Note: * Significant at *p* < 0.05.

**Table 3 ijerph-16-00116-t003:** All-cause mortality by sex and wealth status. Hazard ratio (HR) and 95% confidence intervals (CI) of covariates from multivariate adjusted Cox regression analysis with restricted cubic splines. WHO-INDEPTH SAGE Purworejo longitudinal data (2007–2010).

	Men		Women	
	Poor	Rich	Poor	Rich
N	2521	2608	3141	2727
Death rate/1000	107	92	86	74
Age (years)				
50–59	Ref.	Ref.	Ref.	Ref.
60–69	2.2 (1.4, 3.4) *	3.4 (2.2, 5.3) *	2.6 (1.6, 4.1) *	1.8 (1.1, 2.6) *
70–79	3.8 (2.5, 5.9) *	6.5 (4.1, 10) *	3.8 (2.4, 6.2) *	3.1 (1.9, 4.5) *
80+	5.0 (3.0, 8.1) *	12 (7.1, 20) *	4.8 (2.7, 8.5) *	5.3 (2.7, 8.5) *
Education				
No formal education	1.8 (0.93, 3.6)	1.1 (0.67, 1.8)	2.1 (0.67, 6.7)	1.2 (0.73, 1.9)
≤6 years	1.7 (0.91, 3.3)	1.0 (0.72, 1.4)	1.7 (0.53, 5.4)	0.89 (0.56, 1.4)
>6 years	Ref.	Ref.	Ref.	Ref.
Occupation				
Non-physical labour	Ref.	Ref.	Ref.	Ref.
No occupation	2.1 (0.96, 6.0)	0.81 (0.52, 1.3)	3.9 (1.4, 10) *	2.0 (1.1, 3.8) *
Physical labour	0.87 (0.39, 2.3)	0.46 (0.30, 0.71)	2.2 (0.79, 5.9)	1.4 (0.71, 2.6)
Marital status				
Single/widowed	1.3 (1.0, 1.6)	1.0 (0.71, 1.4)	1.1 (0.86, 1.4)	1.6 (1.2, 2.2) *
Partnership	Ref.	Ref.	Ref.	Ref.
Self-reported chronic disease				
No	Ref.	Ref.	Ref.	Ref.
Yes	1.7 (1.3, 2.2) *	2.4 (1.8, 3.1) *	2.2 (1.7, 2.9) *	1.6 (1.2, 2.2) *
Residence				
Coastal	Ref.	Ref.	Ref.	Ref.
Inland	1.6 (1.2, 2.1) *	1.0 (0.80, 1.4)	1.0 (0.73, 1.4)	0.81 (0.58, 1.1)
Hilly & mountainous	1.2 (0.90, 1.5)	0.81 (0.51, 1.3)	1.1 (0.80, 1.4)	1.3 (0.85, 2.1)

Note: * Significant at *p* < 0.05.

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
