# Peer review of "Waist Circumference and All-Cause Mortality among Older Adults in Rural Indonesia"

_ijerph, 2019, doi:10.3390/ijerph16010116_

Round 1
Reviewer 1 Report
I wish to congratulate the authors for tackling an important public health issue - obesity and mortality in older age. Both are on the increase globally. The article would have been strengthened if the follow up period was longer.
The rationale for the study is well justified. However, the aim was to investigate obesity and mortality in older adults, i.e. 65y and over, but sample had age 50y and over. In the categorisation, the authors grouped sample as 50-59, 60-69y, 70-79y and 80y+. Again, only the last two categories strictly meet the definition of older age as even stated by the authors.
Education categories: what about those who started JHS but did not complete? Please redefine this group
Occupations: driving cannot be said to be physically active
Marital status: You may keep this as it is but the experiences of being single are different from those of separation and divorce widowhood and you should state this in the discussion.
Residence: This not quite clear to me as it appears altitude is the issue. I can understand that this may be important in the particular environment but should be clarified. Most of us are used urban, suburban and rural.
In the results, you cannot have decrease or increase when these are not significant. Please restate this.
Line 207: 9 missing in >90cm.
Tendency towards U-shaped association of waist circumference and mortality in poor women has not been demonstrated. Please revise this point in the article and in the abstract.
You have not really explained the differences in mortality association with waist circumference between men and women. Your explanation is on difference in obesity between men and women in Indonesia.
You also emphasised very much on physical activity and even put it as one of the key words but this not clear from your study. I suggest you remove it from the key word as it will mislead literature searchers.
Author Response
IJERPH-398908
International Journal of Environmental Research and Public Health
Waist Circumference and All-Cause Mortality among Older Adults in Rural Indonesia
Cahya Utamie Pujilestari, Lennarth Nyström, Margareta Norberg, Nawi Ng
Reviewer # 1
I wish to congratulate the authors for tackling an important public health issue - obesity and mortality in older age. Both are on the increase globally. The article would have been strengthened if the follow-up period was longer.
Authors’ response: We thank the reviewer for the detailed and constructive review for this paper. We fully agree with the reviewer on the value of a longer follow-up period. We wish it would be possible for us to secure funding to do longer term follow-up in the future. We have addressed all the reviewer’s comments in detailed as follows.
The rationale for the study is well justified. However, the aim was to investigate obesity and mortality in older adults, i.e. 65y and over, but sample had age 50y and over. In the categorisation, the authors grouped sample as 50-59, 60-69y, 70-79y and 80y+. Again, only the last two categories strictly meet the definition of older age as even stated by the authors.
Authors’ response: It is unclear to us why the reviewer had the impression that our study focused on older adults aged 65 years and over. We have clearly stated in our objective that “This study examined the association between waist circumference and mortality among a population aged 50 years and older in Purworejo District, Central Java, Indonesia.” The term ‘elderly’ and ‘older persons’ are often used to refer to those aged 65 years and older. In this study, however, we defined ‘older persons’ as those aged 50 years and over, as per the World Health Organization’s Study on global AGEing and adult health (WHO SAGE)’s definition (1). We mention this in the background (line 59) and in the methods. The social stratification of age varies across cultures and the specific effects of ageing are influenced by multiple factors such as socioeconomic status and living environment, i.e. urban vs. rural areas (2). In Indonesia, government workers or private company workers retire at age 55 years, while those who live in the villages continue to work beyond pensionable age (e.g. farmers or fishermen) but are less active. The country total life expectancy in 2007 was 67 years, hence we think it is more reasonable to include older people aged 50 years and over, instead of using 65 years and over.
Education categories: what about those who started JHS but did not complete? Please redefine this group
Authors’ response: We have clarified the description about education in the Methods section. In the analysis, we merged those who did not complete junior high school into <6 years education (line 106).
Occupations: driving cannot be said to be physically active
Authors’ response: It is true that in general, drivers are less physically active. However, in our study setting, drivers also do laborious work including lifting or carrying the goods to/from their carriage (e.g. truck of crops). To avoid confusion, we have now used the term “labourer” instead of “driver” (Line 110).
Marital status: You may keep this as it is but the experiences of being single are different from those of separation and divorce widowhood and you should state this in the discussion.
Authors’ response: We categorised marital status based on their social relationships, whether they are living alone or with a spouse, as this might influence their health behaviour at home. We believed that social relationships such as marriage might be related to gender differences in abdominal obesity. In our study, abdominal obesity prevalence was higher among those in partnerships compared to those single/widowed. The mechanisms behind marital status and obesity might be explained through social obligations within marriage. Meals with richer and denser food will be served regularly as part of those traditional obligations of the married couple (3). It can be hypothesised that older couples might have more regular meals, thereby contributing to the prevalence of larger waist measurements among married couples.
Residence: This not quite clear to me as it appears altitude is the issue. I can understand that this may be important in a particular environment but should be clarified. Most of us are used urban, suburban and rural.
Authors’ response: In our study setting, we define the different geographical areas, which also differ socioeconomically into ‘inland’, ‘coastal’, ‘hilly’ and ‘mountainous’. This is not merely an issue of altitude. This has now been clarified in Line 113.
In the results, you cannot have decrease or increase when these are not significant. Please restate this.
Authors’ response: We thank the reviewer for this observation. We have now revised the texts in the results section accordingly (Line 208-221).
Line 207: 9 missing in >2250px.
Authors’ response: This has now been rectified and updated.
Tendency towards U-shaped association of waist circumference and mortality in poor women has not been demonstrated. Please revise this point in the article and in the abstract.
Authors’ response: We thank the reviewer for this comment. We have now rectified the descriptions in both the Abstract and the Results section.
You have not really explained the differences in mortality association with waist circumference between men and women. Your explanation is on difference in obesity between men and women in Indonesia.
You also emphasised very much on physical activity and even put it as one of the keywords, but this not clear from your study. I suggest you remove it from the keyword as it will mislead literature searchers.
Authors’ response: We tried to explain the difference in the association that we believed might be attributable to differences in their death rate and their abdominal obesity prevalence (Line 288-296). In the analysis, we had controlled for other potential confounders such as chronic disease, occupation, etc. Even though we believe that physical activity level can be a potential confounder in the association between waist circumference and mortality, the information about physical activity is not available in our study, hence we discussed the physical activity briefly in the Discussion section. We, however, have never mentioned the term ‘physical activity’ as one of the keywords (Line 32).

Reviewer 2 Report
1. This population-based study aimed to estimate the relationship between gender, waist circumference, economical status, and all-cause mortality.
2. Methods/Results: As the authors mentioned, this study is weak in lack of information regarding specific causes of death, smoking habits, drugs prescribed for participants, and specific comorbid conditions. The follow-up duration was also too short to be of high scientific value. However, it is indeed important to promote prevention strategies beyond wealthier populations who have higher obesity prevalence.
3 As for the issue of reverse causation, I suggest a supplementary sensitivity analysis for Table 2. Please also show the data during 12 and 24 months of follow-up respectively.
Author Response
IJERPH-3989088
International Journal of Environmental Research and Public Health
Waist Circumference and All-Cause Mortality among Older Adults in Rural Indonesia
Cahya Utamie Pujilestari, Lennarth Nyström, Margareta Norberg, Nawi Ng
Reviewer # 2
1. This population-based study aimed to estimate the relationship between gender, waist circumference, economical status, and all-cause mortality.
Authors’ response: We thank the reviewer for the detailed and constructive comments on this paper. We have addressed all the reviewer’s comments as follows.
2. Methods/Results: As the authors mentioned, this study is weak in lack of information regarding specific causes of death, smoking habits, drugs prescribed for participants, and specific comorbid conditions. The follow-up duration was also too short to be of high scientific value. However, it is indeed important to promote prevention strategies beyond wealthier populations who have higher obesity prevalence.
Authors’ response: We thank the reviewer for the critical comment on the duration of the follow-up. The 3-year mortality data was the best available data in this study setting. Even with a follow-up of three years, the data has illustrated the association between mortality and waist circumference. With a longer follow-up and more deaths in the dataset, the insignificant association observed in this study might turn significant. We hope that we could secure more funding to conduct a longer-term follow-up among this cohort of older people in Purworejo District. We believe this study has, however, filled the gaps of knowledge in this research area in a lower middle-income country setting.
3. As for the issue of reverse causation, I suggest a supplementary sensitivity analysis for Table 2. Please also show the data during 12 and 24 months of follow-up respectively.
Authors’ response: Though we appreciate the reviewer’s comment, we do not understand what the reviewer referred to as “reverse causation”. In this study, we focus on waist circumference as an independent variable and mortality as the outcome. We strongly believe that the association between waist circumference was, theoretically, causal and a reserve causation of mortality and waist circumference is implausible. We fully agreed with the reviewer’s critique on short-term follow-up of three years, hence we believe that the suggestion to do sensitivity analysis with 12- and 24-months follow-up period is irrelevant anymore.
During the model building process, we had conducted the AIC (Akaike Information Criterion) test to examine all models (the linear and non-linear model for both full and stratified analyses by sex and wealth status). The non-linear relationships between waist circumference and all-cause mortality were modeled using restricted cubic splines analysis, with 4 degrees of freedom. All the models were adjusted. The model using restricted cubic splines in Cox regression showed the lowest AIC score.
The rationale behind the restricted cubic splines analysis with 4 degrees of freedom is due to the non-linear nature of obesity and mortality relationship, that is tended to be U or J shaped. This approach allowed assessment of the potential non-linear relationship (dose-response relation) in this study. Thus, table 2 is a result from the restricted cubic splines in Cox regression analysis between waist circumference and all-cause mortality with 4 knots chosen at the 5th, 25th, 75th, and 95th percentiles. Table 2 purposefully present the hazard ratios that were extracted in the 5th, 25th, 75th and 95th percentiles of the waist circumferences, where risks were relative to the reference value for each sex. The reference values (men = 90 cm, women = 80 cm) were based on abdominal obesity cut-off value for Asian populations (Line 136-148).
Round 2
Reviewer 2 Report
Please refer to the following articles to help understand why addressing reverse causation issues in observational studies is still a concern.
https://www.ncbi.nlm.nih.gov/pmc/articles/PMC5521974/
https://www.ncbi.nlm.nih.gov/pubmed/27283142
https://www.ncbi.nlm.nih.gov/pubmed/26421898
Author Response
IJERPH-3989088
International Journal of Environmental Research and Public Health
Waist Circumference and All-Cause Mortality among Older Adults in Rural Indonesia
Cahya Utamie Pujilestari, Lennarth Nyström, Margareta Norberg, Nawi Ng
Reviewer#2:
Please refer to the following articles to help understand why addressing reverse causation issues in observational studies is still a concern.
https://www.ncbi.nlm.nih.gov/pmc/articles/PMC5521974/
https://www.ncbi.nlm.nih.gov/pubmed/27283142
https://www.ncbi.nlm.nih.gov/pubmed/26421898
Authors response:
We thank the reviewer for raising the important issue about reverse causation. Chronic disease can lead to underweight, and the effect of overweight is then protective against mortality. We have studied the papers suggested by the reviewer and had conducted sensitivity analyses to test the robustness of the results we reported in the manuscript.
In their paper (https://www.ncbi.nlm.nih.gov/pmc/articles/PMC3806201/), Tobias and Hu suggested two ways to handle reverse causality. The two strategies include: (I) removal of individuals with chronic disease at baseline; and (ii) exclusion of deaths occurring early in the follow-up. We have therefore conducted additional analyses to assess if our results would change if we:
a) as suggested by Tobias and Hu, analyse only individuals without chronic diseases with a follow-up to 36 months (as shown in Table Abelow).
b) as suggested by Tobias and Hu, exclude all deaths occurring in the first 2 years of the study (Table B).
c) as requested by the reviewer, analyse only individuals followed-up to 12 months (Table C)
d) as requested by the reviewer, analyse only individuals followed-up to 24 months (Table D)
Results and interpretations
a) When we controlled for self-reported chronic diseases as a covariatein the regression analysis (Table A),as reported in Table 2in the manuscript, poor men at 5th percentile of waist circumference (HR 2.06, p<0.05) and poor women at 25th percentile of waist circumference (HR 1.39, p<0.05) exhibited a significant increase in mortality risk compared to men with waist circumference of 90 cm or women with waist circumference of 80 cm, the respective reference group of men and women. Though no significant increase of mortality risk was observed among poor men with higher waist circumference, the effect sizes show a decreasing pattern. Such pattern was not observed among rich men. Similar pattern of larger effect sizes was also observed among poor women compared to the rich women, except among rich women with larger waist circumference (95th HR = 1.44; 95% CI = 0.82, 2.5). In the sensitivity analysis when we excluded individuals with self-reported chronic diseasesin the baseline (Table A),we observed similar direction of effect sizes as in the analysis for overall sample, except that all the results become insignificant.
b) When we excluded all deaths happened during the first 23 months of the studyand only include deaths happened during 24-36 months of the study (Table B), poor men at 5th percentile (HR 5.8, p<0.05) and 25th percentile of waist circumference (HR 4.1, p<0.05) exhibited a significant increase in mortality risk compared to men with waist circumference of 90 cm. Poor men with larger waist circumference showed no significant association. No significant association was observed among poor men with larger waist circumference, as well as among rich men and poor and rich women.
c) When we limited our analysis only up to 12 months of follow-up (Table C)as requested by the reviewer, the only significant association with mortality risk was observed among rich women with low waist circumference (5thpercentile HR = 1.9; 95% CI = 1.1, 3.4). When we exclude individuals with self-reported chronic disease in the baseline, no significant association was observed between waist circumference and mortality in all groups of men and women. Rich women with larger waist circumference showed higher HR compared to the poor women, while poor women with low waist circumference showed higher HR compared to the rich women. No clear pattern was shown among men.
d) When we limited our analysis only up to 24 months of follow-up (Table D)as requested by the reviewer, we observed significant higher mortality risk among poor men at 5thpercentile and poor women at 25thpercentile as we observed in the original analysis in Table 2. No significant association was observed among rich men as well as among poor men and women with larger waist circumference. Significant associations with increasing mortality risk were observed among the rich women with low waist circumference (5thpercentile: HR = 1.8 and 25thpercentile: HR = 1.4). In the sub-group analysis where we only include individuals without self-reported chronic disease (N=8,945), no significant association was observed between waist circumference and mortality in all groups of men and women.
Conclusions:
Comparing the original analysis and the additional analyses, the effect size and the trend of the HR observed concurred. Hence, we believe that our analysis based on the full sample followed up to 36 months by controlling chronic disease as a covariate was sensible and provide robust results.
We agree with the reviewer that other factors such as smoking habits, drug prescriptions, and other specific comorbid conditions could also be potential confounders. As the study did not collect this information, we have discussed this as a limitation of this study. A longer follow-up and more observed deaths in the data might turn the insignificant association observed in this study become significant. Nevertheless,the 3-year mortality data was the best available data in this study setting. We believe this study will fill the gaps of knowledge in this research area in a lower middle-income country setting.
Table A. All-cause mortality stratified by sex and wealth status, with follow-up to 36 months.
N | Number of Deaths | Percentiles | ||||||||
5th | 25th | 75th | 95th | |||||||
Waist (cm) | HR (95% CI) | Waist (cm) | HR (95% CI) | Waist (cm) | HR (95% CI) | Waist (cm) | HR (95% CI) | |||
Overall analysis on all samples with self-reported chronic diseases controlled as a covariate in the regression analysis (N=10,997) As shown in Table 2 in the manuscript | ||||||||||
Men | ||||||||||
Poor | 2,521 | 270 | 64 | 2.06 (1.28, 3.31) * | 70 | 1.51 (0.94, 2.41) | 80 | 1.27 (0.83, 1.94) | 88 | 1.06 (0.91, 1.22) |
Rich | 2,608 | 241 | 65 | 1.01 (0.67, 1.54) | 72 | 1.00 (0.69, 1.47) | 83 | 0.92 (0.74, 1.16) | 93 | 1.01 (0.92, 1.12) |
Women | ||||||||||
Poor | 3,141 | 269 | 62 | 1.36 (0.97, 1.90) | 69 | 1.39 (1.06, 1.82) * | 81 | 0.98 (0.93, 1.03) | 92 | 0.89 (0.64, 1.25) |
Rich | 2,727 | 201 | 63 | 1.32 (0.86, 2.03) | 72 | 1.22 (0.89, 1.66) | 85 | 0.98 (0.77, 1.24) | 97 | 1.24 (0.78, 1.98) |
Subgroup analysis after excluding individuals WITH self-reported chronic diseases (N=8,945) Using the same cut-off points as in the overall analysis. | ||||||||||
Men | ||||||||||
Poor | 2,121 | 189 | 64 | 1.56 (0.88, 2.78) | 70 | 1.26 (0.73, 2.17) | 80 | 1.21 (0.67, 2.21) | 88 | 1.14 (0.73, 1.77) |
Rich | 2,114 | 139 | 65 | 0.92 (0. 61, 1.40) | 72 | 0.92 (0.61, 1.40) | 83 | 0.99 (0.73, 1.35) | 93 | 1.01 (0.84, 1.20) |
Women | ||||||||||
Poor | 2,595 | 177 | 62 | 1.33 (0.87, 2.03) | 69 | 1.46 (0.96, 2.22) | 81 | 0.96 (0.88, 1.04) | 92 | 0.67 (0.43, 1.04) |
Rich | 2,115 | 129 | 63 | 0.94 (0.56, 1.57) | 72 | 1.12 (0.84, 1.48) | 85 | 0.95 (0.71, 1.28) | 97 | 1.44 (0.82, 2.52) |
Note: Hazard ratio (HR) and 95% confidence intervals (CI) from multivariate adjusted Cox regression analysis with restricted cubic splinesof waist circumference at 5th, 25th, 75thand 95thpercentiles. WHO-INDEPTH SAGE Purworejo longitudinal data (2007-2010).
Table B. All-cause mortality stratified by sex and wealth status, with follow-up to 36 months and excluding all deaths happened during Month 1-23 of the follow-up
N | Number of Deaths | Percentiles | ||||||||
5th | 25th | 75th | 95th | |||||||
Waist (cm) | HR (95% CI) | Waist (cm) | HR (95% CI) | Waist (cm) | HR (95% CI) | Waist (cm) | HR (95% CI) | |||
Overall analysis on all samples with self-reported chronic diseases controlled as a covariate in the regression analysis (N=10,316) | ||||||||||
Men | ||||||||||
Poor | 2,322 | 71 | 64 | 5.83 (1.41, 24.1) * | 70 | 4.11 (1.06, 16.0) * | 80 | 2.61 (0.55, 12) | 88 | 1.45 (0.51, 4.14) |
Rich | 2,439 | 72 | 65 | 1.74 (0.89, 3.41) | 72 | 1.27 (0.71, 2.29) | 83 | 0.99 (0.77, 1.27) | 93 | 1.53 (0.33, 7.15) |
Women | ||||||||||
Poor | 2,961 | 89 | 62 | 1.20 (0.66, 2.18) | 69 | 1.35 (0.75, 2.44) | 81 | 0.98 (0.79, 1.22) | 92 | 0.43 (0.18, 1.01) |
Rich | 2,594 | 68 | 63 | 0.76 (0.34, 1.70) | 72 | 1.11 (0.76, 1.63) | 85 | 1.14 (0.94, 1.37) | 97 | 3.14 (0.77, 12.8) |
Subgroup analysis after excluding individuals WITH self-reported chronic diseases (N=8,397) Using the same cut-off points as in the overall analysis. | ||||||||||
Men | ||||||||||
Poor | 1,955 | 43 | 64 | 4.16 (0.62, 27.8) | 70 | 4.11 (0.69, 24.4) | 80 | 2.57 (0.27, 24.8) | 88 | 1.21 (0.11, 12.8) |
Rich | 1,996 | 46 | 65 | 1.05 (0.48, 2.26) | 72 | 0.71 (0.35, 1.42) | 83 | 0.88 (0.57, 1.36) | 93 | 1.21 (0.95, 1.56) |
Women | ||||||||||
Poor | 2,442 | 66 | 62 | 1.38 (0.71, 2.69) | 69 | 1.25 (0. 62, 2.50) | 81 | 1.02 (0.87, 1.21) | 92 | 0.56 (0.24, 1.30) |
Rich | 2,004 | 49 | 63 | 0.56 (0.19, 1.63) | 72 | 1.10 (0.70, 1.74) | 85 | 0.78 (0.46, 1.34) | 97 | 1.20 (0.47, 3.04) |
Note: Hazard ratio (HR) and 95% confidence intervals (CI) from multivariate adjusted Cox regression analysis with restricted cubic splinesof waist circumference at 5th, 25th, 75thand 95thpercentiles. WHO-INDEPTH SAGE Purworejo longitudinal data (2007-2010).
Table C. All-cause mortality stratified by sex and wealth status, with follow-up to 12 months.
N | Number of Deaths | Percentiles | ||||||||
5th | 25th | 75th | 95th | |||||||
Waist (cm) | HR (95% CI) | Waist (cm) | HR (95% CI) | Waist (cm) | HR (95% CI) | Waist (cm) | HR (95% CI) | |||
Overall analysis on all samples with self-reported chronic diseases controlled as a covariate in the regression analysis (N=10,997) | ||||||||||
Men | ||||||||||
Poor | 2,521 | 93 | 64 | 1.44 (0.69, 3.00) | 70 | 0.96 (0.47, 1.97) | 80 | 0.94 (0.43, 2.04) | 88 | 1.03 (0.57, 1.87) |
Rich | 2,608 | 95 | 65 | 0.83 (0.38, 1.79) | 72 | 1.18 (0.66, 2.10) | 83 | 1.17 (0.88, 1.56) | 93 | 1.28 (0.23, 7.12) |
Women | ||||||||||
Poor | 3,141 | 103 | 62 | 1.44 (0.84, 2.47) | 69 | 1.38 (0.83, 2.31) | 81 | 0.89 (0.76, 1.04) | 92 | 1.08 (0.53, 2.17) |
Rich | 2,727 | 67 | 63 | 1.90 (1.07, 3.38) * | 72 | 1.32 (0.88, 1.98) | 85 | 1.05 (0.73, 1.52) | 97 | 1.52 (0.70, 3.31) |
Subgroup analysis after excluding individuals WITH self-reported chronic diseases in the baseline (N=8,945) Using the same cut-off points as in the overall analysis. | ||||||||||
Men | ||||||||||
Poor | 2,121 | 68 | 64 | 1.09 (0.45, 2.63) | 70 | 0.82 (0.35, 1.88) | 80 | 1.09 (0.44, 2.67) | 88 | 1.26 (0.64, 2.50) |
Rich | 2,114 | 49 | 65 | 0.88 (0.33, 2.33) | 72 | 1.21 (0.56, 2.59) | 83 | 1.02 (0.55, 1.89) | 93 | 0.90 (0.63, 1.27) |
Women | ||||||||||
Poor | 2,595 | 67 | 62 | 1.71 (0.88, 3.30) | 69 | 1.60 (0.82, 3.12) | 81 | 0.89 (0.80, 0.99) | 92 | 1.04 (0.55, 1.97) |
Rich | 2,115 | 45 | 63 | 1.46 (0.71, 3.02) | 72 | 1.17 (0.72, 1.90) | 85 | 1.12 (0.72, 1.74) | 97 | 1.83 (0.74, 4.50) |
Note: Hazard ratio (HR) and 95% confidence intervals (CI) from multivariate adjusted Cox regression analysis with restricted cubic splinesof waist circumference at 5th, 25th, 75thand 95thpercentiles. WHO-INDEPTH SAGE Purworejo longitudinal data (2007-2010).
Table D. All-cause mortality stratified by sex and wealth status, with follow-up to 24 months.
N | Number of Deaths | Percentiles | ||||||||
5th | 25th | 75th | 95th | |||||||
Waist (cm) | HR (95% CI) | Waist (cm) | HR (95% CI) | Waist (cm) | HR (95% CI) | Waist (cm) | HR (95% CI) | |||
Overall analysis on all samples with self-reported chronic diseases controlled as a covariate in the regression analysis (N=10,997) | ||||||||||
Men | ||||||||||
Poor | 2,521 | 181 | 62 | 2.02 (1.09, 3.73) * | 70 | 1.14 (0.66, 1.98) | 79 | 1.11 (0.60, 2.03) | 87 | 1.16 (0.71, 1.90) |
Rich | 2,608 | 149 | 65 | 0.75 (0.43, 1.28) | 72 | 0.92 (0.61, 1.37) | 84 | 1.01 (0.84, 1.22) | 94 | 1.37 (0.37, 4.99) |
Women | ||||||||||
Poor | 3,141 | 160 | 61 | 1.34 (0.83, 2.17 | 68 | 1.51 (1.00, 2.28) * | 80 | 0.93 (0.86, 1.00) | 93 | 0.74 (0.40, 1.36) |
Rich | 2,727 | 120 | 60 | 1.85 (1.04, 3.30) * | 70 | 1.40 (1.00, 1.95) * | 84 | 1.02 (0.78, 1.32) | 97 | 1.59 (0.91, 2.79) |
Subgroup analysis after excluding individuals WITH self-reported chronic diseases in the baseline (N=8,945) Using the same cut-off points as in the overall analysis. | ||||||||||
Men | ||||||||||
Poor | 2,121 | 137 | 62 | 1.48 (0.74, 2.97) | 70 | 0.98 (0.54, 1.78) | 79 | 1.06 (0.56, 2.03) | 87 | 1.17 (0.68, 2.01) |
Rich | 2,114 | 82 | 66 | 0.76 (0.37, 1.53) | 72 | 1.02 (0.58, 1.80) | 83 | 1.02 (0.64, 1.61) | 91 | 0.92 (0.81, 1.05) |
Women | ||||||||||
Poor | 2,595 | 103 | 61 | 1.34 (0.74, 2.42) | 68 | 1.54 (0.90, 2.61) | 78 | 1.10 (0.92, 1.31) | 89 | 0.78 (0.51, 1.19) |
Rich | 2,115 | 80 | 60 | 1.29 (0.59, 2.82) | 70 | 1.21 (0.79, 1.86) | 85 | 1.04 (0.72, 1.50) | 95 | 1.52 (0.80, 2.90) |
Note: Hazard ratio (HR) and 95% confidence intervals (CI) from multivariate adjusted Cox regression analysis with restricted cubic splinesof waist circumference at 5th, 25th, 75thand 95thpercentiles. WHO-INDEPTH SAGE Purworejo longitudinal data (2007-2010).

Round 3
Reviewer 2 Report
Much better.